# Prognosis and Treatment Outcomes of Bone Metastasis in Gallbladder Adenocarcinoma: A SEER-Based Study

**DOI:** 10.3390/cancers15205055

**Published:** 2023-10-19

**Authors:** Kriti Gera, Doga Kahramangil, Graeme A. Fenton, Daniela Martir, Diana N. Rodriguez, Zohaib Ijaz, Rick Y. Lin, Sherise C. Rogers, Brian H. Ramnaraign, Thomas J. George, Young-Rock Hong, Steven J. Hughes, Ibrahim Nassour, Ilyas Sahin

**Affiliations:** 1Department of Medicine, University of Florida College of Medicine, Gainesville, FL 32610, USA; kriti.gera@medicine.ufl.edu (K.G.); diana45@ufl.edu (D.N.R.); zohaib.ijaz@medicine.ufl.edu (Z.I.); ricklin@medicine.ufl.edu (R.Y.L.); 2Division of Hematology and Oncology, Department of Medicine, University of Florida College of Medicine, Gainesville, FL 32610, USA; dogakahramangilb@ufl.edu (D.K.); sherise.rogers@medicine.ufl.edu (S.C.R.); brian.ramnaraign@medicine.ufl.edu (B.H.R.); thom.george@medicine.ufl.edu (T.J.G.); 3University of Florida Health Cancer Center, Gainesville, FL 32610, USA; youngrock.h@phhp.ufl.edu; 4University of Florida College of Medicine, Gainesville, FL 32610, USA; graeme.fenton@ufl.edu; 5Department of Radiation Oncology, University of Florida College of Medicine, Gainesville, FL 32610, USA; dmar0013@shands.ufl.edu; 6Department of Health Services Research, Management and Policy, University of Florida College of Medicine, Gainesville, FL 32610, USA; 7Division of Surgical Oncology, Department of Surgery, University of Florida College of Medicine, Gainesville, FL 32610, USA; steven.hughes@surgery.ufl.edu (S.J.H.); ibrahim.nassour@surgery.ufl.edu (I.N.)

**Keywords:** biliary tract cancer, cholangiocarcinoma, bone metastasis

## Abstract

**Simple Summary:**

Gallbladder cancer (GBC) is the most frequently diagnosed biliary tract cancer, associated with a poor prognosis due to its aggressive nature and insidious onset. Using data from the SEER database between the years 2011 and 2020, we analyzed the demographic factors and outcomes of 2724 patients who were diagnosed with metastatic gallbladder adenocarcinoma, the most common subtype of gallbladder carcinoma. The objective of this study was to investigate the demographic characteristics and assess the impact of bone metastasis on survival outcomes, as well as the effects of chemotherapy utilization in the presence of bone metastasis. Our results showed that patients with bone metastasis had significantly reduced overall survival rates compared to those without bone metastasis, particularly at younger ages. Furthermore, the utilization of chemotherapy was associated with improved survival outcomes in patients with bone metastasis.

**Abstract:**

Background: Gallbladder carcinoma (GBC) is a rare, aggressive malignancy comprising 0.5% of gastrointestinal cancers. It has poor survival outcomes due to its insidious onset, lack of standardized screening, and limited therapies. Advanced-stage diagnosis with liver, lymph node, and peritoneal metastasis is common, while bone metastasis is rare. The knowledge on bone metastasis in GBC is limited to case reports and small series, and its clinical significance is largely unexplored. Methods: The study extracted the demographic and clinical variables of patients with metastatic (M1) gallbladder adenocarcinoma from the Surveillance, Epidemiology, and End Results (SEER) database between 2011 and 2020. Descriptive statistics were used to analyze the demographic characteristics. The multivariate Cox regression analysis was used to calculate the hazard ratio. The overall survival (OS) was assessed using the Kaplan–Meier method, and the log-rank test was utilized to compare the survival between the groups. Results: A total of 2724 patients were included in the study. A total of 69% of the patients were female, and the median age was 68 (range 24–90+). A total of 7.4% of the patients had bone metastasis on diagnosis. The multivariate Cox analysis identified bone metastasis as an independent mortality risk factor in metastatic GBC (HR 1.50, *p* < 0.001). The patients were divided into two age groups: a younger age group (18–74 years) and an older age group (75+ years). In the younger group, the median OS with and without bone metastasis was 3 and 5 months, respectively (*p* < 0.0001). In the older age group, there was no significant difference in the OS between the patients with and without bone metastasis (*p* = 0.35). In the younger group who were treated with chemotherapy, the patients with bone metastasis had a significantly worse OS (median OS 5 months vs. 8 months, *p* < 0.0001). In the untreated group, the patients with bone metastasis in the younger age group had a significantly worse OS (median OS 1 month vs. 2 months, *p* = 0.014). In the patients with bone metastasis, those who did not receive chemotherapy had a significantly worse OS than those who were treated with chemotherapy in both age groups (younger age group: median OS 1 month vs. 5 months, *p* < 0.0001 and older age group: median OS 1 month vs. 5 months, *p* = 0.041). Conclusions: Our findings suggest that the presence of bone metastasis in gallbladder adenocarcinoma is an independent prognostic factor associated with unfavorable survival outcomes in the younger age group (18–74 years). However, in the older age group (75+ years), the presence of bone metastasis did not impact the survival. Treatment with chemotherapy was associated with extended survival in all patients. Thus, early detection and aggressive management of bone metastasis, including the consideration of chemotherapy, may be crucial in improving the OS and quality of life for individuals with gallbladder adenocarcinoma.

## 1. Introduction

Gallbladder cancer (GBC) is an aggressive biliary tract malignancy characterized by a dismal prognosis. Estimates by the American Cancer Society suggest 12,220 new cases of GBC will be diagnosed in 2023, resulting in 4510 associated deaths [1]. Although GBC is a relatively uncommon cancer in the United States, accounting for roughly 0.5% of all gastrointestinal malignancies, it demonstrates considerable geographic variability with particularly high rates of incidence observed in northern India, Chile, and Japan [2]. Epidemiologically, there is a well-documented history of increased rates of GBC incidence among women, Hispanics, and older age groups [3].

GBC is the most common form of biliary tract cancer (BTC)—a group that also includes the intrahepatic, perihilar, and extrahepatic forms of cholangiocarcinoma—and has the shortest associated median survival duration of the group [4]. This high mortality is attributed to the aggressive nature of the malignancy, insidious onset of the disease, absence of effective screening protocols, and a limited range of available therapeutic interventions. As such, the majority of patients present with metastatic disease upon diagnosis. The most common sites of metastasis are the liver, regional lymph nodes, and peritoneum, with only 20% of cases characterized by non-abdominal metastasis, primarily to the lung and brain [5]. Bone metastasis is the least common manifestation of GBC; however, the incidence can be as high as 10%, as evidenced in an autopsy series [6]. Given the lack of standardized screening measures for detecting bone metastasis, a comprehensive understanding of the incidence and prognostic significance of skeletal involvement in GBC is lacking. 

Currently, surgical resection is the only means of treatment with curative intent for local and locoregionally advanced disease; however, only 10% of patients qualify as surgical candidates and the likelihood of cure from surgery remains low [7]. As such, combinatorial treatment regimens incorporating chemotherapy, targeted therapies, and immunotherapies are actively being explored to extend survival and quality of life [8,9]. Despite several reported cases and series in the literature, population-level data on treatment outcomes and prognostic evaluation of GBC remains limited, punctuated by anecdotal experiences [6,10,11].

The poor prognosis of GBC can be attributed to its insidious onset without early symptom manifestation, the absence of effective screening, and the aggressive nature of the disease, which leads to rapid progression [12,13]. This often leads to a delayed diagnosis at moderate and advanced stages, where surgery, the only potentially curative treatment for GBC, is no longer feasible [14,15]. Furthermore, beyond the already established poor prognosis of GBC, the presence of isolated bone metastasis in GBC is associated with an even more unfavorable outcome when compared to isolated lung and distant lymph node metastasis [16]. However, there is a paucity of literature exploring bone metastasis in gallbladder cancers, with the majority of the existing data being derived from case reports and small series [6,10,11]. To address this research gap, we used the Surveillance, Epidemiology, and End Results (SEER) database, a large cancer registry, to conduct a comprehensive analysis of bone metastasis in patients with gallbladder adenocarcinoma, the most common histologic subtype of the gallbladder cancers [15]. Our study aimed to investigate the impact of skeletal involvement on survival outcomes and assess the potential clinical benefits of the utilization of chemotherapy in this setting. The findings from this study will help provide new insight into this highly fatal manifestation and interventional strategies that could improve detection and survival.

## 2. Methods

### 2.1. Data Source

The 22 Population-Based Registries Research Plus Data in the National Cancer Institute SEER database from 2000 to 2020 (November 2022 submission) was accessed for this study [17]. The SEER Program gathers cancer incidence and survival data from population-based cancer registries that represent roughly 48 percent of the U.S. population. SEER-22 registry includes the following sites: Alaska, Arizona, Atlanta (Georgia), Connecticut, Detroit (Michigan), Hawaii, Iowa, New Mexico, Rural Georgia, San Francisco—Oakland (California), San Jose—Monterey (California), Seattle Puget Sound (Washington), Utah, Kentucky, Los Angeles, Louisiana, New Jersey, Greater Georgia, Greater California, Idaho, Illinois, Massachusetts, New York, Texas, and Utah [17].

### 2.2. Patient Selection

The primary goal was to analyze treatment outcomes. In 2010, gemcitabine plus cisplatin became the standard systemic treatment [18], so to avoid discrepancy in treatment choices and ensure adequate follow-up for overall survival, the period of 2011–2020 was chosen for inclusion in the study. To identify GBC patients, topography code C23.0 was used. Inclusion criteria consisted of only one primary site, ICD-O-Histology code of 8140–8389 (adenocarcinoma), complete survival, metastasis, and treatment (chemotherapy) information. Only patients at M1 stage based on TNM staging categorized as a subgroup in IVB of AJCC (8th edition) staging were included in this study.

### 2.3. Study Variables

The demographic characteristics of GBC patients extracted from SEER included age at diagnosis, sex, race, ethnicity, survival status, and survival time. The race/ethnicity variable included Hispanic (all races), Non-Hispanic American Indian/Alaska Native, Non-Hispanic Asian or Pacific Islander, Non-Hispanic Black, Non-Hispanic White and Non-Hispanic unknown race. Patients were divided into two groups based on receipt of treatment (treated with chemotherapy or not) to analyze prognostic significance of bone metastasis in these groups. Patients were also divided into two groups based on presence of bone metastasis (yes and no) to reflect the effect of treatment in patients with bone metastasis. All the patients reported between 2011 and 2019 in the SEER database were included in this study, and the target event was death by any cause. The survival time was the time interval since the date of diagnosis to either death date or last follow-up date for those still alive in 2020. Subjects who were still alive by the end of follow-up in 2020 were considered censored for the event.

### 2.4. Statistical Analysis

Descriptive statistics were used to summarize baseline demographic characteristics.

Survival data was calculated from date of diagnosis to last follow-up or death from any cause. The multivariate Cox regression analysis was used to calculate hazard ratio. Overall survival was estimated using Kaplan–Meier method and the log-rank test was used to compare survival amongst groups. A *p*-value < 0.05 was considered statistically significant. All statistical analyses were performed using R software version 4.3.1.

## 3. Results

### 3.1. Patient Characteristics

A total of 2724 patients satisfied the inclusion criteria and were included in the analysis (Figure 1). The distribution of site-specific metastasis was as follows: 1.1% (*n* = 30) to the brain, 15% (*n* = 396) to the lung, 68% (*n* = 1863) to the liver, and 7.4% (*n* = 202) to the bone. The median age was 68 years (24, 90+) and 69% (*n* = 1874) were females. A total of 50% of patients were non-Hispanic white, and 25% belonged to the non-Hispanic group. A total of 55.7% (1518) patients with metastatic GBC received chemotherapy treatment. The baseline characteristics of the study subjects are outlined in Table 1.

A total of 202 (7.4%) patients had bone metastasis at the time of diagnosis with or without the presence of metastasis to other sites. Among these patients, 6.9% (*n* = 14), 33.1% (*n* = 67), and 60.4% (*n* = 122) had concurrent brain, lung, and liver metastasis, respectively on diagnosis. The median age was 65 years (25, 90+). A total of 62% patients were female and 53% were Non-Hispanic White. A total of 58% (117) patients received chemotherapy. The baseline characteristics of these patients is outlined in Table 2.

### 3.2. Factors Impacting Survival

The multivariate Cox analysis showed that the factors associated with an increased risk of mortality included an age of more than 75 years (HR 1.20, CI 1.09–1.32, *p* < 0.001), bone metastasis (HR 1.50, CI 1.29–1.75, *p* < 0.001), and black race (HR 1.25, CI 1.11–1.41, *p*-value < 0.001). The factors that were associated with an improved OS included Hispanic race (HR 0.86, CI 0.77–0.95, *p* = 0.002), and treatment with chemotherapy (HR 0.33, CI 0.31–0.36, *p* < 0.001) (Figure 2).

To mitigate the confounding effect of age, the patients were stratified into two distinct age groups: those younger than 75 years (18–74 years) and those older than 75 years.

### 3.3. Prognostic Significance and Survival

A separate univariate survival analysis of the stratified groups (younger age group [18–74 years] and older age group [75+ years]) was performed to evaluate the OS based on the different covariates. The median OS of all the patients with metastatic GBC (M1 stage) was 4 months. In the younger age group, the median OS for patients with and without bone metastasis was 3 and 6 months, respectively (*p* < 0.0001). In contrast, there was no significant difference in the OS between patients with and without bone metastasis in the older group (median OS 3 months for both, *p* = 0.32) (Figure 3).

Similarly, in the treatment group, there was significantly worse survival in the patients with bone metastasis versus the patients without bone metastasis in the younger group (median OS of 5 months versus 8 months, *p* < 0.0001). In contrast, in the older age group, there was no significant difference in the OS between patients with and without bone metastasis, with a median OS of 3 months and 7 months, respectively (*p* = 0.13) (Figure 4).

In the untreated group, there was a significantly worse survival in the younger age group (18–74 years) with a median OS of 1 month for patients with bone metastasis and 2 months for patients without bone metastasis (*p* = 0.014). In contrast, there was no significant difference in the OS between patients with and without bone metastasis in the older age group (75+ years) (median OS of 1 month in both, *p* = 0.67) (Appendix A).

### 3.4. Treatment Outcomes

#### 3.4.1. Metastatic GBC

In both age groups, the patients who received chemotherapy treatment showed a significantly improved OS compared to the patients who did not receive chemotherapy treatment. In the younger age group (18–74 years), the median OS was 2 months for patients who did not receive chemotherapy treatment and 8 months for patients who received chemotherapy treatment (*p* < 0.0001). Similarly, in the older age group (75+ years), the median OS was 1 month for patients who did not receive chemotherapy treatment and 7 months for patients who received chemotherapy treatment (*p* < 0.0001) (Appendix A).

#### 3.4.2. Metastatic GBC with Bone Metastasis

Regardless of the age group, the median OS for patients who did and did not receive chemotherapy was 5 months and 1 month, respectively. The survival was significantly worse in patients who did not receive chemotherapy (*p* < 0.0001 for younger group and *p* = 0.041 for older group) (Figure 5).

## 4. Discussion

GBC stands as the most prevalent and aggressive form of BTC, affecting 2.2 per 100,000 people worldwide every year [19,20,21]. The poor prognosis of GBC can be attributed to its insidious onset without early symptom manifestation, the absence of effective screening, and the aggressive nature of the disease resulting in rapid progression [12,13]. This often leads to a delayed diagnosis at advanced stages where surgery, the only potentially curative treatment for GBC, is no longer indicated [14,15]. Furthermore, beyond the already established poor prognosis of GBC, the presence of bone metastasis in GBC is suggested to be associated with an even more unfavorable outcome when compared to isolated lung and distant lymph node metastasis [16]. There is limited research on bone metastases in gallbladder cancer, with most of the available information coming from case reports and small studies. To address this research gap, we conducted a comprehensive analysis of bone metastasis in gallbladder adenocarcinoma patients using the SEER database. Our study found that patients with gallbladder adenocarcinoma have a lower overall survival rate when bone metastasis is also present.

Older age and female sex are among the most well-established risk factors for the development of GBC. The average age observed in the literature for GBC is 65 years [22], and female patients demonstrate an approximately two-fold higher incidence of GBC worldwide [19,20,23,24]. Consistent with these findings, our cohort revealed a median age of 68 years, with the majority (69%) being female. The higher incidence of GBC in women is due to the higher prevalence of gallstones and the influence of female sex hormones [25]. In their study utilizing the SEER data from 2001 to 2012, Jaruvongvanicha et al. demonstrated that Hispanic patients had the highest incidence rate of GBC among the ethnicity groups, while Black patients exhibited a 1.6 times higher incidence rate compared to White patients [20]. On the other hand, using SEER data between 2004 and 2015, Jiang et al. demonstrated the predominance of White patients in a surgical cohort of GBC patients [26]. It is important to note, however, that there may have been sampling issues in that surgical study [26]. In our cohort, where only patients with metastatic (M1) GBC were included, non-Hispanic White patients constituted the majority (50%), while Hispanic patients still accounted for 25% of the cohort. The observed difference in the representation among minority ethnic groups in the surgical setting compared to the incidence of GBC reflected in the general public could be attributed to variations in healthcare access; however, a thorough analysis spanning the years from 2012 to 2020, considering genetic, environmental, and social factors, examining incidence trends, and addressing possible selection biases, is necessary to comprehensively understand any demographic discrepancies.

Our research findings indicate that both an advanced age and being of Black race are associated with unfavorable survival outcomes. These findings are in line with previous population-based studies that have also reported increased mortality among older individuals, often attributed to factors like compromised immune function, reduced treatment tolerance, and diminished life expectancy. A SEER study encompassing all patients diagnosed with gallbladder adenocarcinoma between 2004 and 2015 also identified a link between Black race and advanced age with higher mortality rates [27]. However, when investigating racial disparities in GBC, another SEER study found no significant differences in the stage at diagnosis among racial groups, though it did reveal that Black individuals were less likely to receive curative surgical treatments [28]. Similar disparities in the receipt of curative surgery have been observed in studies involving other malignancies, such as pancreatic cancer and hepatocellular cancers [29,30]. The potential racial differences in the treatment response and healthcare access have not been thoroughly explored in the context of gallbladder cancer. Consequently, our results emphasize the necessity for future research to delve into the factors contributing to poorer outcomes in this particular patient demographic.

Our study demonstrated significantly reduced overall survival rates in patients with bone metastasis in the younger age group with metastatic GBC. Specifically, those without bone metastasis exhibited a median survival of 6 months, while those with bone metastasis had a median survival of 3 months. Intriguingly, in the older population, the presence of bone metastasis did not impact survival rates in metastatic GBC with a median survival of 3 months regardless of bone metastasis. This may be explained by the inherent survival expectancy in this age group. Our cohort demonstrated a shorter OS compared to a multicentric retrospective study conducted by Santini et al., which showed a median overall survival of 10.9 months for GBC with bone metastasis [11]. Despite the lower overall survival rates, our findings align with the existing literature that has highlighted the poor prognosis associated with the presence of isolated bone metastasis for lung, prostate, and gallbladder cancers [31,32,33,34]. It should be noted, however, that while the skeletal system is the most common site of metastatic disease of breast and prostate cancer, recent research does suggest that bone metastasis is associated with a better median OS compared to other sites of involvement [35,36]. Likewise, the presence of isolated bone metastasis was associated with a statistically significant increase in the OS compared to isolated lung or liver metastasis in patients with metastatic upper tract urothelial carcinoma [37]. Nonetheless, the presence of bone metastasis emerges as an independent and adverse prognostic factor that should be considered in the diagnostic workup of GBC. Notably, our data demonstrate that the significant reduction in overall survival for GBC extends to the presence of bone metastasis in general, beyond the isolated involvement of the skeletal system alone. As bone scans are not currently a routine investigation during diagnostic evaluation [38], our results suggest that the increased use of bone scans to assess for skeletal involvement following initial diagnosis may be warranted to better assess the prognosis and appropriate therapy. While overall an uncommon presentation, elevated alkaline phosphatase levels may not serve as a reliable laboratory sign of bone involvement given the biliary nature of the disease in question.

Systemic chemotherapy is currently the preferred standard of care for non-resectable gallbladder cancers (locally advanced, recurrent, and metastatic) [39]. A combination of gemcitabine and cisplatin serves as the first line regimen for advanced and/or metastatic BTC—including GBC—after the ABC-02 trial in 2010 demonstrated a significant increase in the median overall survival (to 11.7 months) among the patients receiving the combined treatment, compared to those receiving gemcitabine as a single agent (8.1 months) [18]. The recent phase 3 trials, TOPAZ-1 and KEYNOTE-966, demonstrated extended survival rates by incorporating immunotherapy (durvalumab or pembrolizumab) to the gemcitabine and cisplatin regimen [40,41]. Yet, the influence of these immunotherapies on bone metastasis continues to be indeterminate.

While clinical trials evaluating the chemotherapy response in BTC have traditionally pooled GBC and cholangiocarcinoma together given the low rate of incidence, a recent meta-analysis assessing 58 studies found that the progression-free and overall survival did not differ between the varying forms of BTC [42]. In their SEER analysis between 2010 and 2016, Yang et al. found that the use of chemotherapy was associated with enhanced OS and cancer-specific survival (CSS) among patients with metastatic gallbladder adenocarcinoma who have isolated liver or distant lymph node metastasis compared to those with isolated bone metastasis [34]. In our cohort, treatment with chemotherapy was associated with improved overall survival outcomes in GBC with bone metastasis. It is important to note that although the use of chemotherapy was linked to improved survival outcomes, the SEER data lacks variables pertaining to patients’ physical health, such as comorbidities. The patients in the chemotherapy treatment group were an average of 7 years younger than those who did not receive chemotherapy. This age discrepancy may correspond to an observed reduction in comorbidities for patients in the treatment group compared to those who did not receive treatment. The variation in the survival rates observed may be attributed to these factors, as the administration of chemotherapy typically requires a higher overall baseline health status. Future analysis will require consideration of these comorbidities to better assess the true efficacy of chemotherapy in patients with metastatic GBC. Intravenous bisphosphonate therapy is typically indicated for patients with bone metastasis from other solid tumors in an effort to reduce skeletal related events (e.g., fractures, pain, etc.). Our data suggest that patients with GBC may not live long enough to benefit from such interventions.

Some studies have suggested a survival benefit from radical surgery for stage IV disease [43]; however, conflicting evidence has shown increased surgical complications and mortality rates [44] along with a lack of improvement in disease progression in larger scale research [45,46]. Furthermore, Yang et al. also showed that surgery was not helpful when metastasis to bone or lungs is present [34]. The assessment of surgical outcomes and associated prognosis was not within the scope of this study and thus was not evaluated.

Due to the retrospective nature of the SEER data and incomplete reporting by some states and cancer centers, there are some limitations to our study. It is important to consider the potential for selection bias and confounding factors and interpret the data with caution. Moreover, certain variables that may have influenced the progression of the disease, such as the performance status and comorbidities, were not reported in the database, potentially limiting our full understanding of their impact.

## 5. Conclusions

In conclusion, this study represents the largest study to date exploring the course and clinical impact of bone metastasis, as well as the utilization of chemotherapy, in the setting of metastatic GBC. Our findings revealed that the presence of bone metastasis is associated with inferior overall survival. Although chemotherapy treatment prolongs survival, the overall prognosis for patients with bone metastasis remains grim, even with implementation of therapeutic interventions. Given the infrequent presentation of patients with bone metastasis, but significant association with poor prognosis, bone metastasis should be considered as a stratification variable for future clinical trials. In light of the recent approval of immunotherapy for BTC, evaluating the impact of immunotherapy on survival rates in GBC patients with bone metastasis is a relevant follow-up question. Moreover, given that at least one potentially actionable molecular alteration is found in half of all GBC patients [47], future studies could explore how the application of molecular testing and subsequent targeted therapy options might improve survival rates for these individuals. Prospective studies are required to investigate the distinct characteristics of bone metastasis and its treatment, as these findings have the potential to provide valuable insights for clinical decision-making.

## Figures and Tables

**Figure 1 cancers-15-05055-f001:**
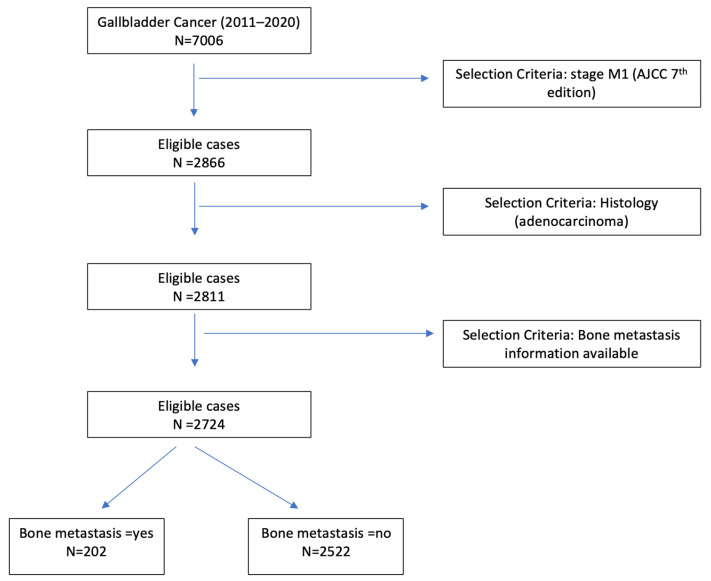
Patient selection flowchart.

**Figure 2 cancers-15-05055-f002:**
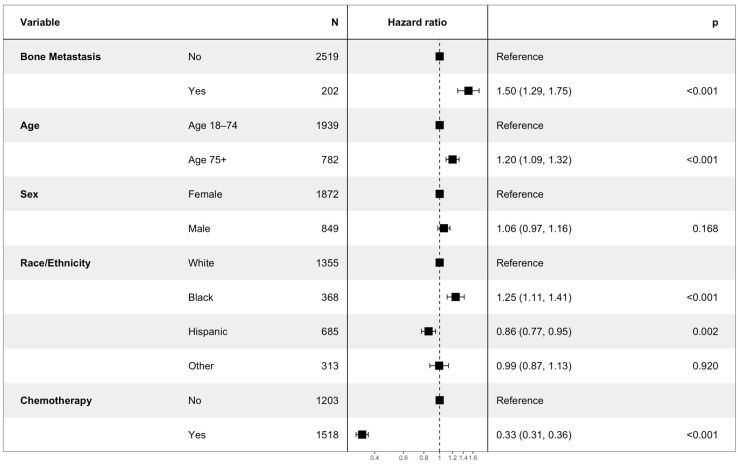
Hazard ratio using Cox proportional-hazards analysis. Hazard ratios > 1 indicate an increased risk of mortality, while hazard ratios < 1 indicate decreased risk of mortality. The *p*-value for each variable is depicted on the right side of the figure.

**Figure 3 cancers-15-05055-f003:**
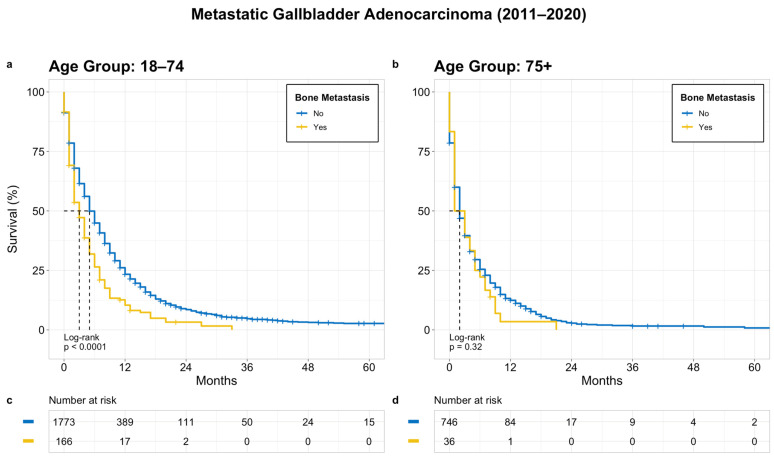
OS in patients with metastatic GB adenocarcinoma based on bone metastasis; (**a**) age 18–74 years, (**b**) age 75+ years, (**c**) and (**d**) denotes patients at risk at a given time point.

**Figure 4 cancers-15-05055-f004:**
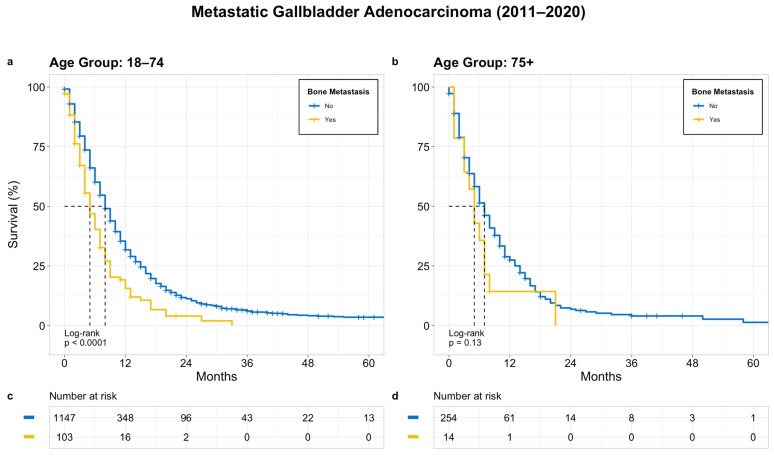
OS in patients with treated metastatic GB adenocarcinoma based on bone metastasis; (**a**) age 18–74 years, (**b**) age 75+ years, (**c**) and (**d**) denotes patients at risk at a given time point.

**Figure 5 cancers-15-05055-f005:**
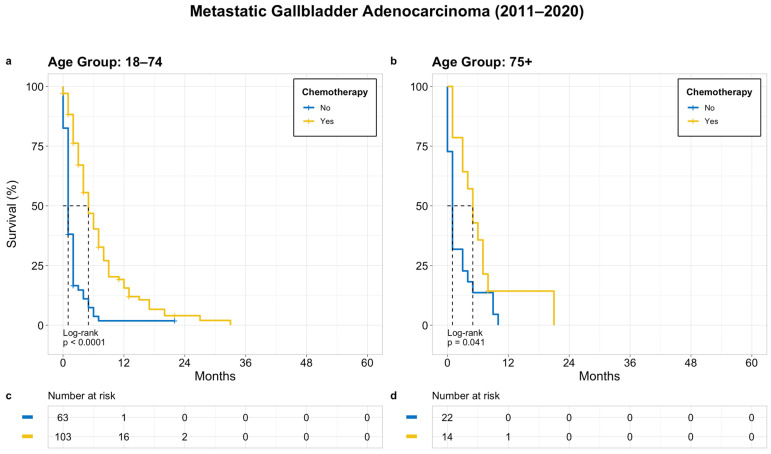
OS in patients with bone metastasis based on receipt of chemotherapy; (**a**) age 18–74 years, (**b**) age 75+ years, (**c**) and (**d**) denotes patients at risk at a given time point.

**Table 1 cancers-15-05055-t001:** Baseline characteristics of patients with metastatic gallbladder adenocarcinoma divided based on receipt of chemotherapy treatment.

		Chemotherapy Treatment	
Variable	Overall, *n* = 2724 ^1^	No, *n* = 1206 ^1^	Yes, *n* = 1518 ^1^
Age at Diagnosis			
Median (IQR)	68.0 (60.0, 76.0)	72.0 (64.0, 80.0)	65.0 (57.0, 72.0)
Range	24.0, 90.0	31.0, 90.0	24.0, 89.0
Sex			
Female	1874 (69%)	845 (70%)	1029 (68%)
Male	850 (31%)	361 (30%)	489 (32%)
Race/Ethnicity			
Hispanic (All Races)	685 (25%)	339 (28%)	346 (23%)
Non-Hispanic American Indian/Alaska Native	31 (1.1%)	18 (1.5%)	13 (0.9%)
Non-Hispanic Asian or Pacific Islander	279 (10%)	121 (10%)	158 (10%)
Non-Hispanic Black	369 (14%)	152 (13%)	217 (14%)
Non-Hispanic Unknown Race	3 (0.1%)	0 (0%)	3 (0.2%)
Non-Hispanic White	1357 (50%)	576 (48%)	781 (51%)
Bone Metastasis	202 (7.4%)	85 (7.0%)	117 (7.7%)
Year of diagnosis			
2011	226 (8.3%)	106 (8.8%)	120 (7.9%)
2012	230 (8.4%)	100 (8.3%)	130 (8.6%)
2013	242 (8.9%)	107 (8.9%)	135 (8.9%)
2014	251 (9.2%)	116 (9.6%)	135 (8.9%)
2015	256 (9.4%)	109 (9.0%)	147 (9.7%)
2016	301 (11%)	141 (12%)	160 (11%)
2017	306 (11%)	131 (11%)	175 (12%)
2018	284 (10%)	119 (9.9%)	165 (11%)
2019	325 (12%)	153 (13%)	172 (11%)
2020	303 (11%)	124 (10%)	179 (12%)
Survival months			
Median (IQR)	4 (1, 9)	1 (0, 3)	7 (3, 12)
Range	0, 107	0, 79	0, 107
Histology			
8140/3: Adenocarcinoma, NOS	2347 (86%)	1040 (86%)	1307 (86%)
8144/3: Adenocarcinoma, intestinal type	12 (0.4%)	4 (0.3%)	8 (0.5%)
8160/3: Cholangiocarcinoma	331 (12%)	146 (12%)	185 (12%)
8211/3: Tubular adenocarcinoma	1 (<0.1%)	1 (<0.1%)	0 (0%)
8255/3: Adenocarcinoma with mixed subtypes	14 (0.5%)	7 (0.6%)	7 (0.5%)
8260/3: Papillary adenocarcinoma, NOS	12 (0.4%)	6 (0.5%)	6 (0.4%)
8263/3: Adenocarcinoma in tubulovillous adenoma	1 (<0.1%)	0 (0%)	1 (<0.1%)
8310/3: Clear cell adenocarcinoma, NOS	6 (0.2%)	2 (0.2%)	4 (0.3%)

^1^ *n* (%).

**Table 2 cancers-15-05055-t002:** Baseline characteristics of patients with metastatic GB adenocarcinoma when stratified based on presence of bone metastasis.

		Bone Metastasis	
Variable	Overall, *n* = 2724 ^1^	No, *n* = 2522 ^1^	Yes, *n* = 202 ^1^
Age at Diagnosis			
Median (IQR)	68.0 (60.0, 76.0)	68.0 (60.0, 76.0)	65.0 (57.0, 72.0)
Range	24.0, 90.0	24.0, 90.0	25.0, 90.0
Sex			
Female	1874 (69%)	1749 (69%)	125 (62%)
Male	850 (31%)	773 (31%)	77 (38%)
Race/Ethnicity			
Hispanic (All Races)	685 (25%)	640 (25%)	45 (22%)
Non-Hispanic American Indian/Alaska Native	31 (1.1%)	30 (1.2%)	1 (0.5%)
Non-Hispanic Asian or Pacific Islander	279 (10%)	262 (10%)	17 (8.4%)
Non-Hispanic Black	369 (14%)	338 (13%)	31 (15%)
Non-Hispanic Unknown Race	3 (0.1%)	3 (0.1%)	0 (0%)
Non-Hispanic White	1357 (50%)	1249 (50%)	108 (53%)
Chemotherapy	1518 (56%)	1401 (56%)	117 (58%)
Year of diagnosis			
2011	226 (8.3%)	217 (8.6%)	9 (4.5%)
2012	230 (8.4%)	213 (8.4%)	17 (8.4%)
2013	242 (8.9%)	222 (8.8%)	20 (9.9%)
2014	251 (9.2%)	233 (9.2%)	18 (8.9%)
2015	256 (9.4%)	239 (9.5%)	17 (8.4%)
2016	301 (11%)	271 (11%)	30 (15%)
2017	306 (11%)	280 (11%)	26 (13%)
2018	284 (10%)	267 (11%)	17 (8.4%)
2019	325 (12%)	304 (12%)	21 (10%)
2020	303 (11%)	276 (11%)	27 (13%)
Survival months			
Median (IQR)	4 (1, 9)	4 (1, 9)	3 (1, 6)
Range	0, 107	0, 107	0, 33
Histology			
8140/3: Adenocarcinoma, NOS	2347 (86%)	2179 (86%)	168 (83%)
8144/3: Adenocarcinoma, intestinal type	12 (0.4%)	12 (0.5%)	0 (0%)
8160/3: Cholangiocarcinoma	331 (12%)	301 (12%)	30 (15%)
8211/3: Tubular adenocarcinoma	1 (<0.1%)	1 (<0.1%)	0 (0%)
8255/3: Adenocarcinoma with mixed subtypes	14 (0.5%)	12 (0.5%)	2 (1.0%)
8260/3: Papillary adenocarcinoma, NOS	12 (0.4%)	11 (0.4%)	1 (0.5%)
8263/3: Adenocarcinoma in tubulovillous adenoma	1 (<0.1%)	1 (<0.1%)	0 (0%)
8310/3: Clear cell adenocarcinoma, NOS	6 (0.2%)	5 (0.2%)	1 (0.5%)

^1^ *n* (%).

## Data Availability

The data that support the findings of this study are publicly available from https://seer.cancer.gov/.

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
