# Peer review of "Prognosis and Treatment Outcomes of Bone Metastasis in Gallbladder Adenocarcinoma: A SEER-Based Study"

_cancers, 2023, doi:10.3390/cancers15205055_

Round 1

Reviewer 1 Report

Knowledge on bone metastasis in gallbladder cancer (GBC) is limited and its clinical significance is lacking.

This article is a comprehensive analysis of bone metastasis in gallbladder adenocarcinoma using the SEER database.

The objective of the study is to investigate the demographic characteristics and assess the impact of bone metastasis on survival outcomes, as well as the effects of chemotherapy use in the presence of bone metastasis.

Methods include extraction of demographic and clinical variables of patients with metastatic (M1) GB adenocarcinoma from SEER database between 2011-2020. Descriptive statistics were used to analyse demographic characteristics. Overall Survival (OS) was assessed using the Kaplan-Meier method and comparison of survival between groups was performed.

The findings revealed that the presence of bone metastasis in GB adenocarcinoma is associated with inferior OS in the younger age group (18-74 years). However, presence of bone metastasis did not impact survival in the older age group (75+ years). Treatment with chemotherapy was associated with extended survival in all patients.

The study has also some limitations; the SEER database lacks certain variables pertaining to patients’ physical health, such as performance status and comorbidities which were not reported in the database (due to the retrospective nature of data), potentially limiting the full understanding of their impact. Future analysis will need consideration of these comorbidities to better assess the true efficacy of chemotherapy in patients with metastatic GBC.

Results from this study provide new insights into GB adenocarcinoma with bone metastases and interventional strategies that could improve detection and survival.

Minor corrections to make: 

3. Results, line 159-160: “A total of 202 (7.4%) patients had bone metastasis at diagnosis. Median age was 68 years (25,90+)” 68 changed to 65 years (25,90+) (Table 2)

Line 134 and line 193, typo error.

Author Response

We value the reviewers' constructive feedback and appreciate their positive comments. Below, you'll find our detailed responses to each of the reviewers' suggestions. All the edits have been highlighted in the manuscript.

Reviewer 1:

Authors: We appreciate the positive feedback from reviewer 1 and opportunity to enhance the quality of work.

Comment #1

Results, line 159-160: “A total of 202 (7.4%) patients had bone metastasis at diagnosis. Median age was 68 years (25,90+)” 68 changed to 65 years (25,90+) (Table 2)

Authors: We apologize for the error and appreciate the reviewer for addressing the error.

Comment #2: Line 134 and line 193, typo error.

Authors: We apologize for the typo error, and it has been addressed.

Reviewer 2 Report

I read with great interest the paper on Prognosis and Treatment Outcomes of Bone Metastasis in 2 Gallbladder Adenocarcinoma.

The SEER db was explored.

The sample size is adequate to draw reliable conclusions.

The authors included only M1 patients, however, it is not clear which were the other metastatic sites other than bone. It would be interesting to know the distribution of other sites. 

Did authors include patients with single site met or were patients with multiple met also included? Please specify this in the methods and in the results.

Overall, the analysis is well performed and the results may be fluently read.

Lastly, please rely on https://doi.org/10.3390/jcm11185310, for differences with other metastatic tumors in the discussion section.

Author Response

We value the reviewers' constructive feedback and appreciate their positive comments. Below, you'll find our detailed responses to each of the reviewers' suggestions. All the edits have been highlighted in the manuscript.

Reviewer 2:

We appreciate the positive feedback from reviewer 2 and opportunity to enhance the quality of work.

Comment #1: The authors included only M1 patients, however, it is not clear which were the other metastatic sites other than bone. It would be interesting to know the distribution of other sites. 

Authors: The text has been amended to include the suggested information, “The distribution of site-specific metastasis was as follows: 1.1% (n=30) to the brain, 15% (n=396) to the lung, 68% (n=1,863) to the liver, and 7.4% (n=202) to the bone.” (Lines 155-156).

Comments #2: Did authors include patients with single site met or were patients with multiple met also included? Please specify this in the methods and in the results.

Authors: The text has been amended to include the suggested information, “A total of 202 (7.4%) patients had bone metastasis at the time of diagnosis with or without presence of metastasis to other sites. Among these patients, 6.9% (n=14), 33.1% (n=67) and 60.4% (n=122) had concurrent brain, lung and liver metastasis respectively on diagnosis” (Lines 161-164).

Comment #3: Lastly, please rely on https://doi.org/10.3390/jcm11185310, for differences with other metastatic tumors in the discussion section.

Authors: The text has been amended to include the suggested information, “Likewise, the presence of isolated bone metastasis was associated with a statistically significant increase in OS compared to isolated lung or liver metastasis in patients with metastatic upper tract urethral carcinoma”. (Lines 303-305). (Reference has been included).